# Calming Hungarian Grey Cattle in Headlocks Using Processed Nasal Vocalization of a Mother Cow

**DOI:** 10.3390/ani14010135

**Published:** 2023-12-30

**Authors:** Ádám Lenner, Zoltán Lajos Papp, Csaba Szabó, István Komlósi

**Affiliations:** 1Doctoral School of Animal Science, University of Debrecen, 4032 Debrecen, Hungary; 2Department of Computer Science, Faculty of Informatics, University of Debrecen, 4032 Debrecen, Hungary; pappzolaj@gmail.com; 3Department of Animal Nutrition and Physiology, Faculty of Agricultural and Food Sciences and Environmental Management, University of Debrecen, 4032 Debrecen, Hungary; szabo.csaba@agr.unideb.hu; 4Department of Animal Husbandry, Faculty of Agricultural and Food Sciences and Environmental Management, University of Debrecen, 4032 Debrecen, Hungary; komlosi@agr.unideb.hu

**Keywords:** stress, temperament, cattle headlock, vocalization, heart rate variability

## Abstract

**Simple Summary:**

Two of the fundamental pillars of animal welfare are stress-free management and treatment. The extensive farming of cattle involves minimal human contact. However, regular veterinary treatments and vaccinations require restraining cattle with headlocks in crates in order to ensure the safety of workers. Since these animals are usually kept in herds and mainly graze freely, this procedure is a high-stress situation for them. Animals can vocalize a number of sounds as part of their behavior. For example, dams have a special nasal vocalization toward their calves with a calming effect. We hypothesized that, by removing noises, selecting the most representative part, and creating a repeated sound, this kind of nasal vocalization would have a calming effect on cattle restrained in headlocks. Our results show that the played processed nasal vocalization of a mother cow reduced the stress experienced by cattle during the test.

**Abstract:**

Sound analysis is an important field of research for improving precision livestock farming systems. If the information carried by livestock sounds is interpreted correctly, it could be used to improve management and welfare assessment in this field. Therefore, we hypothesized that the nasal vocalization of a mother cow could have a calming effect on conspecifics. The nasal vocalization in our study was recorded from a mother cow (not part of the test herd) while it was licking its day-old calf. The raw sound was analyzed, cleaned from noises, and the most representative vocalization was lengthened to two minutes. Thirty cows having calves were randomly selected from eighty Hungarian grey cattle cows. Two test days were selected, one week apart; the weather circumstances in both days were similar. The herd was collected in a paddock, and the test site (a restraining crate with a headlock) was 21 m away from them. The cows from the herd were gently moved to the restraining crate, and, after the installation of the headlock, Polar^®^ heart rate monitors were fixed on the animals. The recording of the RR intervals was carried out for two minutes. On day one of the test, the processed nasal sound was played to every second cow during the heart rate monitoring. When the sound ended, the heart rate monitor was removed. On test day two, the sound and no sound treatments were switched among the participating cows. At the end of the measurement, the headlock was opened, letting the animals out voluntarily, and a flight test was performed along a 5 m distance. The time needed to pass the 5 m length was measured with a stopwatch and divided by the distance. The RR intervals were analyzed with the Kubios HRV Standard (ver. 3.5.0) software. The following data were recorded for the entire measurement: average and maximum heart rate; SD1 and SD2; pNN50; VLF, LF, and HF. The quasi-periodic signal detected in the sound analyses can hardly be heard, even when it is enhanced to the maximum. This can be considered a vibration probably caused by the basis of articulation, such as a vibration of the tongue, for example. The SD2/SD1 ratio (0.97 vs. 1.07 for the animals having no sound and sound played, respectively, *p* = 0.0110) and the flight speed (0.92 vs. 1.08 s/m for the animals having no sound and sound played, respectively, *p* = 0.0409) indicate that the sound treatment had a calming effect on the restrained cows. The day of the test did not influence any of the measured parameters; therefore, no effect of the routine was observed. The yes–no sequence of the sound treatment significantly reduced the pNN50 and flight speed values, suggesting a somewhat more positive association with the headlock and the effectiveness of the processed nasal sound. In conclusion, we have demonstrated that, by means of sound analyses, not only information about individuals and the herd can be gathered but that, with proper processing, the sound obtained can be used to improve animal welfare.

## 1. Introduction

Consumers’ animal welfare demands have become important factors in the beef industry [1]. In addition to the importance of meat quality, consumers are increasingly concerned with humane treatment and animal welfare [1]. In modern farming, there is an increased demand for innovative tools that collect and analyze information pertaining to livestock as well as individual animals. Mother cows move away from their conspecifics only during calving; in other situations, they experience separation as stress. However, during individual veterinary procedures, cattle often need to be separated from their conspecifics. As a result of separation, cattle show significant physiological changes, including increases in heart rate, saliva cortisol levels, and frequency of bladder and bowel movements [2] as well as changes in the activity of said movement [2,3]. Sudden noises, stimuli, and procedures are accompanied by stress for the animals, which has evolved from anti-predator responses [4,5]. Stress negatively affects production; hence, farmers need to reduce stress caused by farming technologies [6,7]. A flighty temperament affects body mass growth, meat quality, and state of health in cattle [8]. The more active and stressed an individual is, the worse its feed conversion [9,10,11], meat quality [12,13,14], immune functions [15], time they spend lying down [16], and weight growth are [9,16,17,18,19], and the more animals repeatedly becoming agitated during restraining need to be culled [20]. Nervous cattle have higher basal concentrations of catecholamines (adrenaline, noradrenaline, and dopamine) and cortisol in their blood than those of calm cattle [21,22,23]. This explains why it is necessary to keep not only farming conditions stress-free but compulsory animal health procedures too. More temperamental cattle are prone to injuries during treatments, and they also show a higher incidence of dark cutting (DFD) [12,24]. They also pose a danger to the veterinary surgeons attending to them as well as to the equipment used during their treatment [25]. In contrast, calm cattle have a higher average daily gain (ADG) [9,26,27], and the fertilization of mother cows is easier [28,29]. Tense and nervous cattle are much more sensitive to the activities that cause them stress [30], including being kept in a cattle handling crate during handling. Temperament is a complex phenomenon which is repeated in particular situations; thus, significant differences can be observed between individual cattle [31]. The temperament of cattle can be measured in practice using flight speed (FS) [32,33,34], one of the most reliable methods [16,22,35,36,37]. Flight speed measures the time over which an individual traverses a particular passageway. More temperamental individuals cover the given distance or passageway faster, while calmer ones take longer to cover the predetermined distance [32,38].

Heart rate (HR) has been used as a measure to determine stress levels in various animals, including cattle [39,40], since the 1950s [41]. Recently, there has been increased interest in heart rate variability (HRV), which provides detailed and immediate information about stress levels [42]. The time that lapses between two heartbeats is called an R-R interval, whose variability can be used to calculate several parameters [43,44]. Heart rate variability is caused by the fluctuation of the balance between sympathetic and parasympathetic tones [45,46]. The sympathetic response increases heart rate [47,48,49], while the parasympathetic one decreases it [50]. Results of recent research in veterinary science and in the physiology of behavior have shown that HRV is a precise tool for examining autonomous nervous system activity [51,52,53]. It is also suitable for gaining information about the well-being of the fetus in the last 100 days of pregnancy [54]. HRV parameters can show immediate changes; thus, we can gain information about stress that lasts for a short time [55,56,57]. Results of research carried out in cattle reveal that R-R intervals are a good indicator to determine stress from being kept in a cattle handling crate [58].

A positive experience is an indicator of good animal welfare [59,60]. Thus, human behavior is also important as it can determine an animal’s response [61,62]. Since stroking, soft human speech [63], and low-frequency sounds [64] have a positive effect on cattle, we assumed that a mother cow’s nasal vocalization would also have a calming effect [65] on cattle in headlocks, knowing that they experience stress when restrained [66]. Cattle vocalization has been analyzed using several approaches in the past, but the nasal sound type that we selected has not yet been analyzed with the methods we applied. Other researchers analyzed oral vocalizations triggered by separation, estrus, or other kinds of stress [65,67,68]. This study aimed to reduce the stress of Hungarian grey cattle caused by being kept in a crate using the pre-recorded sound of a mother cow calling its calf. 

## 2. Materials and Methods

### 2.1. Experimental Animals and Housing

The research was conducted on the Szamárhát Hungarian grey cattle farm owned by the Tiszatáj Public Foundation. On the farm, extensive animal husbandry takes place, but the animals spend the winter period in sheds. Every year they are subjected to compulsory blood tests and various vaccinations. On those occasions, crates with headlocks are used to separate and restrain the animals from their usual surroundings and the other animals.

At the time of the experiment, 80 cows were rearing calves out of the 190 cows in the cattle herd, and 30 were selected randomly from those reared calves to take part in the test. Cross-over designs were used [69], which created two equal-numbered groups (15 individuals in each group) of mother cows.

### 2.2. Nasal Sound Recording and Analysis

The nasal vocalization emitted by a mother cow (9 years old, third calving, not in the herd during the trial) to her day old calf while licking it was previously recorded using a Sony IC Recorder AX412F (Sony Europe B.V., Budapest, Hungary), analyzed, and processed using the Signal Processing Toolbox program. The Signal Processing Toolbox is a collection of tools built on the MATLAB^®^ (R2021b) numeric computing environment. The toolbox supports a wide range of signal-processing operations, from waveform generation to filter design and implementation, parametric modeling, and spectral analysis. The Signal Processing Toolbox™ provides functions and apps to manage, analyze, preprocess, and extract features from uniformly and nonuniformly sampled signals. The toolbox includes tools for filter design and analysis, resampling, smoothing, detrending, and power spectrum estimation. The Signal Analyzer app can be used for visualizing and processing signals simultaneously in the time, frequency, and time–frequency domains. The most representative vocalization was finally selected and was lengthened to 2 min by repeating the same basic sound using the GoldWave 6.28 software (see Appendix A).

### 2.3. Experimental Procedure

Two days were selected for testing, one week apart. On day one, the processed sound was played to half of the group (every second cow), and, on day two, the treatment was switched. The examined cows entered the corridor on their own; consequently, their order was random, as the people driving the cattle did not influence the order. The people carrying out the measurement did not move or talk nor did they have any contact with the animals during the two minutes of the measurement. As there is no instrument specifically designed to measure heart rate variation in cattle, it is common practice to use different types of Polar^®^ heart rate monitors (Polar Ltd., New York, NY, USA) [44,63,70,71,72]. The measuring unit consists of an elastic belt with two electrodes and a detector (H10). Electrode gel was used to improve conductivity through thick fur [73,74,75]. Furthermore, in our trial, we found it necessary to fix the heart rate monitor with a strong strap [74,76,77] in order to improve connection. The Polar H10 heart rate monitor was applied to the intervertebral part between the third and sixth vertebrae of the mother cows in a crate using bungee cords, as previously carried out by others [78].

The vocalization was played using a JBL Xtreme 2 Bluetooth loudspeaker after the mother cows had been placed in crates with the heartrate monitors attached to them. As soon as the strap was applied and heart rate monitoring started, the sound was played. Before the measurement, the group was located 21 m away from the mother cows being observed behind the shed so they could not hear the vocalization nor see the animals being observed. The waiting cows were kept in an area of 490 square meters. The volume of the recorded vocalization was not measured, but the cattle in the waiting group showed no sign of hearing it. 

### 2.4. HRV Analyses

The heart rate and RR intervals were measured for 2 min immediately after the heart rate monitor was applied and during the playing of the processed nasal vocalization. The people performing the examination and the tools they used were the same on the two test days. The RR data files were processed using the Kubios HRV Standard (ver. 3.5.0) software. The following data were recorded for the entire measurement: average and maximum heart rate; SD1 (Poincaré plot standard deviation perpendicular the line of identity, ms) and SD2 (Poincaré plot standard deviation along the line of identity, ms); RMSSD (root mean square of successive RR interval differences); pNN50 (percentage of successive RR intervals that differ by more than 50 ms, %); VLF, LF, and HF (oscillations into very-low-frequency (0.0033–0.04 Hz), low-frequency (0.04–0.15 Hz), and high-frequency (0.15–0.4 Hz) bands, %). A total of 23 individual cows produced measurable and analyzable data on both test days, and only these data were used for our statistical analyses. The animals were not completely still in the headlocks, which presumably resulted in the electrodes’ displacement; furthermore, the cows’ thick hair may have blocked signal detection even though electrode gel had been used to improve connectivity, resulting in unanalyzable RR data.

### 2.5. Temperament Assessment

At the end of the 2 min, the monitor was removed, and the flight speed (FS) method was used to detect temperament. The mother cows started to walk on their own and proceeded down the corridor. They had to walk along a pre-painted line, which was 5 m long, between the starting point and the finish line. A stopwatch was used to measure the time. To obtain the FS, the time needed to cover the given distance was divided by the length of the distance (s/m) [79].

### 2.6. Meteorological Effects

During the examination, air temperature, air pressure, and air humidity data were recorded using a Technoline Ltd. WS-2800IT instrument (Gżira, Malta). During the two days of data recording, the meteorological circumstances were almost the same. The relative pressure (hPa) was 1026.8 hPa on the 26 May 2021 and 1027.4 hPa on the 2 June 2021. The average air temperature was 21.58 degrees Celsius on the 26 May 2021 and 22.30 degrees Celsius on the 2 June 2021. The air humidity was 46.75% on the 26 May 2021 and 41.00% on the 2 June 2021.

### 2.7. Statistical Analysis

The analysis of the heart rate variability and flight speed data was performed with SAS On Demand For Academics (SAS Institute Inc., Cary, NC, USA) using the GLM procedure. The effect of sound hearing, test day, and sequence of treatments in the change-over design were tested. Group differences were tested using Tukey’s test. Significance was set at *p* ≤ 0.05.

## 3. Results and Discussion

### 3.1. Analysis of the Used Sound

Like other sound lengths in the literature [80], the sound in this study is a close-to-one-second, quasi-stationary, and quasi-periodic signal with a short rise and a quick fall (Figure 1). Its spectrogram (Matlab, Signal Processing Toolbox, Spectrogram using short-time Fourier transform) is shown in Figure 2. The magnified image of the low-frequency band of the spectrogram reveals periodicity and stationarity. There, it can also be seen that there are many higher harmonics of the fundamental tone, with barely decreasing intensities, which means that this sound causes a sharp and crisply defined sense of pitch.

Figure 1 shows that there is a 40 ms delay before the vocal cords become tense. The energy spectrum of this 40 ms interval is shown in Figure 3; it is almost a brown noise, and the fall of the energy spectrum is approximately 1/x^2^. In this spectrum, the effect of the basis of articulation (pharyngeal, oral, and nasal cavities) on the spectrum is also represented.

A magnified image of the spectrum from the middle section (Figure 4) shows that, based on the 10th higher harmonic of the fundamental tone, the fundamental frequency is 767/10 = 76.7 Hz, which is almost the same as other results in the literature [65]. Besides the fundamental tone, there is another one that is deeper and more intense. Their preliminary frequency ratio is 5:16. The lower sound is close to the border of audibility; it is almost a rattle. The transitions at the beginning and at the end of the signal are free from impulse-like noise. Only the fundamental frequency, which appears at the beginning of the signal and disappears towards its end, is worth noting. We defined them as the maxima of their cepstra (Matlab, Signal Processing Toolbox, Real Cepstrum). Based on the cepstrum, the estimated frequency is 44,100/610 = 72.3 Hz at the beginning and 44,100/584 = 75.5 Hz at the end (Figure 5).

Consequently, the middle of the signal can be considered quasi-periodic. Above, we calculated the average of the fundamental frequency of the signal (76.7 Hz) with the method described in the study. To calculate the local values, we used autocorrelation. We estimated the fundamental period to be the maximum values of the autocorrelation (Figure 6).

This shows that there is fluctuation, even if to a minimal extent. This fluctuation can be suppressed by adding the right polynomial. Assuming that this polynomial function is a good approximation of the function which defines the fundamental period, the main component triggering the sense of pitch can be eliminated from the signal as follows: we need to reduce each value of the signal by the value that the signal had a period earlier. Then, by doing so, we will obtain a signal that is the sum of a very-low-frequency, quasi-periodic signal and noise. This is depicted by the spectrum of this composite signal in Figure 7.

We can assume that a value in the noise component of the original signal and the value a period earlier are random values and are independent of each other; therefore, the energy of the noise in the new signal will be approximately double the energy of the noise in the original signal, while the energy of the quasi-periodic part—which is almost sinusoidal—depends on the frequency ratios. If we suppose that the values of a sinusoidal signal with period *T* are reduced by the previous value of sinusoidal Δ*T*, we also obtain a sinusoidal signal with period T but shifted, and its energy is its (2-2 cos(2π ∆*T*/*T*))-fold. The two average periods are the following: Δ*T =* 1/76.7 and Hz = 13.04 ms. And, as it is in the spectrum, *T* = 1/25 and Hz = 40 ms. From here, the energy multiplier is circa 2.92. By considering the two peak points of the spectrum as parts of the quasi-periodic component and the rest as parts of the noise, we obtain two signals after correcting the distortion. This can be compared with the original signal (Figure 8). The quasi-periodic signal can hardly be heard even when it is enhanced to the maximum. This can be considered a vibration probably caused by the basis of articulation, more exactly by a vibration of the tongue, for example.

Speech signals are usually characterized by their formant structure (Matlab, Signal Processing Toolbox, Formant Estimation with LPC Coefficients). Formant structures are functions of time; they vary from time to time. Formants of a moment, more precisely of its environment, are the set of the characteristic peaks of the energy spectrum. A formant is a pair of two frequency values, of which one is the position of the peak (local maximum) on the frequency axis—this is the so-called formant frequency. The other one is the formant width, which refers to the width of the environment where the energy spectrum decreases by 3 dBel, and, because 10lg (2) = 3.010, it is also the environment where the energy spectrum is reduced by half. The lower this value is, the more characteristic the peak is. Formant structures contain formants in ascending order of formant frequencies. In the case of quasi-periodic speech signals, the F_0_ fundamental frequency is usually added to the structure as the 0 index element. Figure 9 shows the energy spectrum of a section of the signal, its approach with linear predictor coefficients, and its formant structure. The formant’s width can also be seen in the figure (horizontal lines).

If this is also calculated for successive, time-interleaved details of the signal, we obtain Figure 10 for the formant frequencies. A separate figure shows the frequency function of the quasi-periodic section of the signal, i.e., the temporal change in F_0_. The marked band represents a semitone interval. Here, the width of the formants is not displayed, only the characteristic formant frequencies are. Because the greatest formant is very close to half of the sampling frequency, that formant can even be attributed to the weak design of the filter used in the recording device. Speech sounds are characterized by the first three formants. The first formants of English vowels are 200–1300 Hz, while their second formants are in the range of 500–3500 Hz. The 4th, 5th, etc., formants characterize the speaker. The individuality of an specific member of a given cattle herd is represented by formants above 5000 Hz. The main Matlab functions for the estimation of formant structure are LPC (linear predictor coefficients), roots, and sort.

In many cases, the analysis of human hearing is based on the ensemble of critical bands, and the mel scale-based cepstrum (Matlab, Audio Toolbox, Mel Frequency Cepstral Coefficients) has a special role in it. It can be assumed that the hearing organ of cattle differs from that of humans only in its dimensions. If we divide the wider hearing range of cattle into critical bands, there are expected to be one or two more critical bands than in the case of humans. In cattle, this solid cepstrum can also be the starting point for examining individuality.

### 3.2. Heart Rate Variability and Flight Speed

In accordance with the results found in the literature [46,50,81], our data also support that a reduction in stress can be detected via changes in the ratio of SD2/SD1. In our study, the SD2/SD1 ratio verified the sound effect significantly (*p* = 0.0110), i.e., the cows to which the nasal vocalization had been played were calmer (Table 1). Human studies have also confirmed that the SD2/SD1 ratio is lower when a subject does not experience stress [81]. No significant difference was found for the heart rate, VLF, HF, and LF parameters in terms of sound effects (*p* > 0.05).

The two test days took place one week apart, which does not qualify as routine because mother cows need 10 min long repetitions twice a day for 5 days to build a routine [82]. This is in line with our results, as no significant test day effect was found.

To assess temperament, a flight speed (FS) measure was used, in which the time needed to travel 1 m (s/m) was calculated. Some authors used distances such as 1.7, 1.83, or 2 m [13,22,83], which are very short in our opinion. However, we had a 5 m corridor available for the measurement. Our results also confirm that more temperamental animals cover the predetermined distance in a shorter time [32,38]. Cattle that had not been exposed to the processed nasal vocalization of the mother cow were more nervous due to the stress caused by the restraint; hence, as soon as their movement ceased to be limited, they covered the 5 m distance faster than those that had been played the nasal vocalization (*p* = 0.0409).

As half of the mother cows were tested in the sequence of sound played and not played (YN) and the other group had the opposite order (NY), we had the chance to test whether this sequence had a significant effect on any of the parameters. Quite interestingly, pNN50 (*p* = 0.0396), LF (*p* = 0.0176), and flight speed (*p* = 0.0199) showed a positive response. While pNN50 and flight speed show that the cows in the YN sequence were calmer (suggesting a somewhat more positive association with the headlock), the LF ratio was higher in the NY sequence of cows (a possible signal of a negative association with the headlock and a less effective sound treatment). These data are controversial and call for further studies to understand the governing mechanisms at play but prove, at the same time, that HRV indices are sensitive parameters. 

Heart rate variability (HRV) parameters are very useful in detecting diseases and assessing the risk factors of diseases as well as in the classification of disease severity and stress in humans [81,84]. HRV variables have been shown to change in response to stress induced using various methods [57]. The observed sensitivity of HRV parameters in human studies has inspired great amounts of research on animals to study their response to exercise, well-being, behavioral disorders, housing and management problems, and temperament and stress [85,86]. Cattle (dairy cows and calves mainly) have also been subject to HRV analyses [42,44]. The researchers in these studies have examined mainly the SDNN and RMSSD from the time domain and the LF, HF, and LF/HF ratio from the frequency domain (but various lower and upper frequency limits are used to define these parameters) [42]. In these studies, between lactating and non-lactating cows no significant differences were observed in HRV in both the time and frequency domains [44]. Calves expressed different RMSSD, HF (normalized), and LF/HF ratios to external stress (ambient temperature > 20 °C and insect harassment), and dairy cows responded similarly to chronic stress (lameness) [72]. However, in our experiment, none of these parameters were found to respond to external stress (restrain with headlock). In other experiments, the SD2/SD1 ratio was lower in lame dairy cows (chronic stress) [72], similarly to our results. Furthermore, Melillo et al. [81] also found a lower SD2/SD1 ratio in mentally stressed (exam) students. These results indicate that, regardless of the stress source, lower SD2/SD1 ratios can be expected.

Although other authors have suggested that cattle separated from their conspecifics vocalize more [2,3,87] in order to restore contact with the herd, all the individuals in our study remained quiet during data collection. Bioacoustics studies have primarily focused on dairy cattle and, within them, on weaning and sounds emitted in response to weaning. Currently, an analysis of the effects of cattle’s nasal and oral sounds is ongoing. We have found no other examples in the literature using the method that we used to analyze the pre-recorded nasal vocalization. The length of the sound used in our experiment agrees with the lengths found in the other literature data reviewed [67]. Although cattle have distinct calls [65], the nasal vocalization of the particular individual that we used in our experiment had a calming effect on all the cattle tested. The individual features of sounds develop with age [65]; hence, we find it important to mention that the nasal vocalization used in our research belonged to a 10-year-old dam. In the case of the Nigerian dwarf goat (*Capra hircus*), past research proved that it self-calmed by emitting a low-frequency sound [64], which led us to assume that this could be true for dams too. Our assumption seemed verifiable because bulls also produce this nasal sound as some kind of courtship behavior while guarding estrous cows [88], and they produce a regular, low-frequency “mm” sound before feeding, which is also nasal in nature and has positive connotations [67].

## 4. Conclusions

We were able to demonstrate the effect of hearing a cow’s nasal vocalization on the SD2/SD1 ratio and flight speed measures of Hungarian grey cattle. A mother cow’s nasal vocalization can be used in precision animal husbandry during treatments and procedures. Our results can be used to improve animal welfare and productivity alike.

## Figures and Tables

**Figure 1 animals-14-00135-f001:**
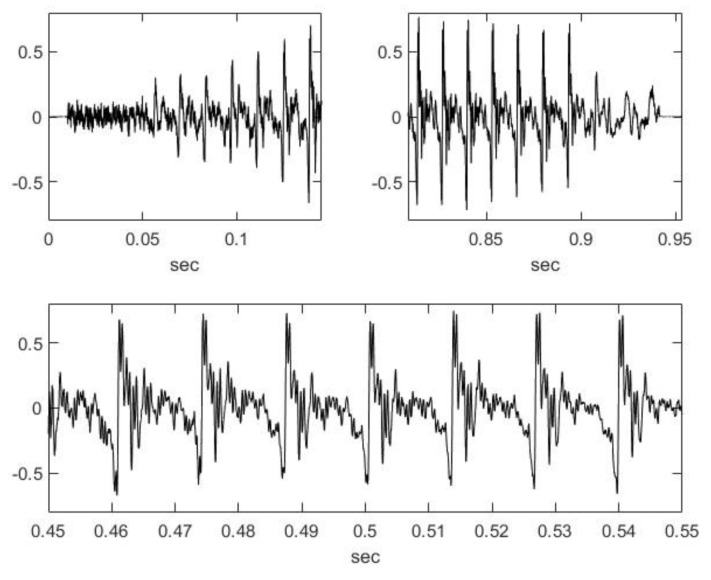
Sample’s beginning, ending, and middle.

**Figure 2 animals-14-00135-f002:**
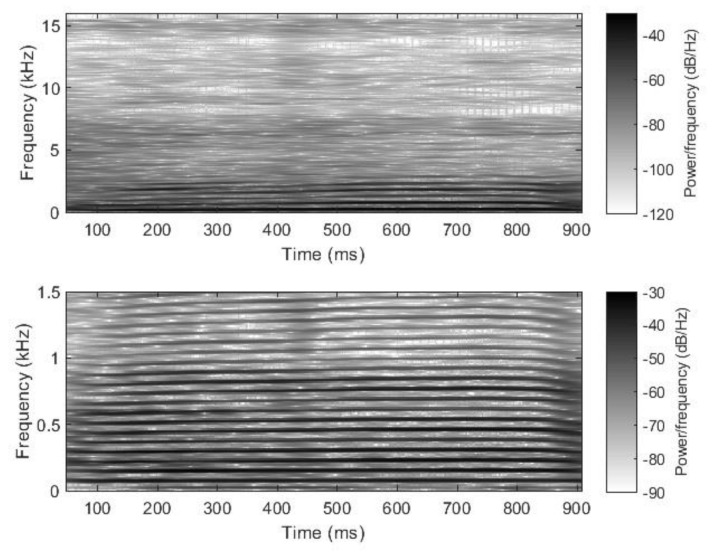
Spectrogram of the entire sample.

**Figure 3 animals-14-00135-f003:**
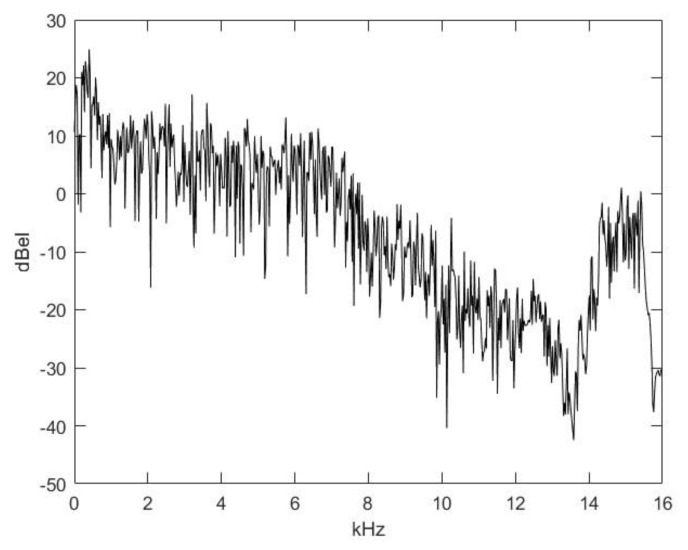
Spectrogram of the introductory noise.

**Figure 4 animals-14-00135-f004:**
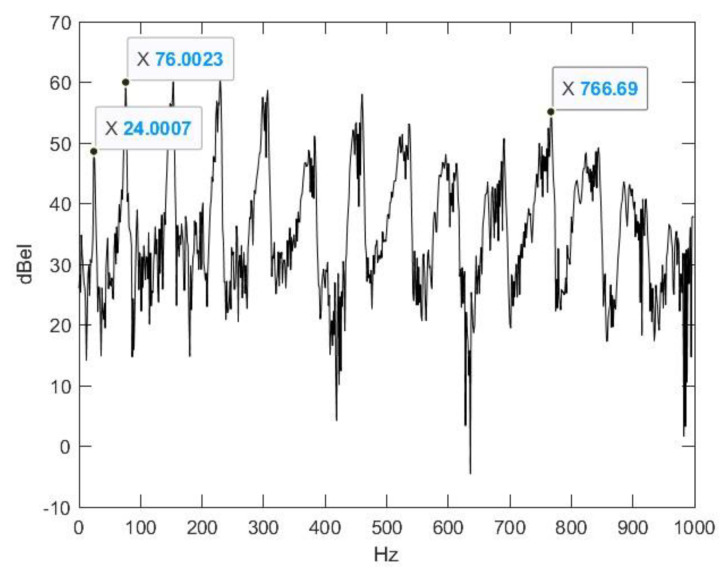
Magnified energy spectrum of the middle section of the signal.

**Figure 5 animals-14-00135-f005:**
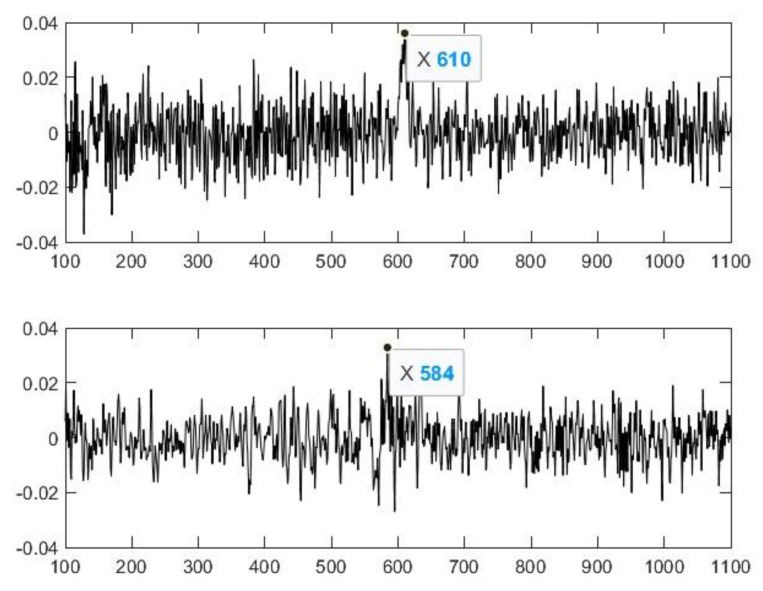
The two cepstra.

**Figure 6 animals-14-00135-f006:**
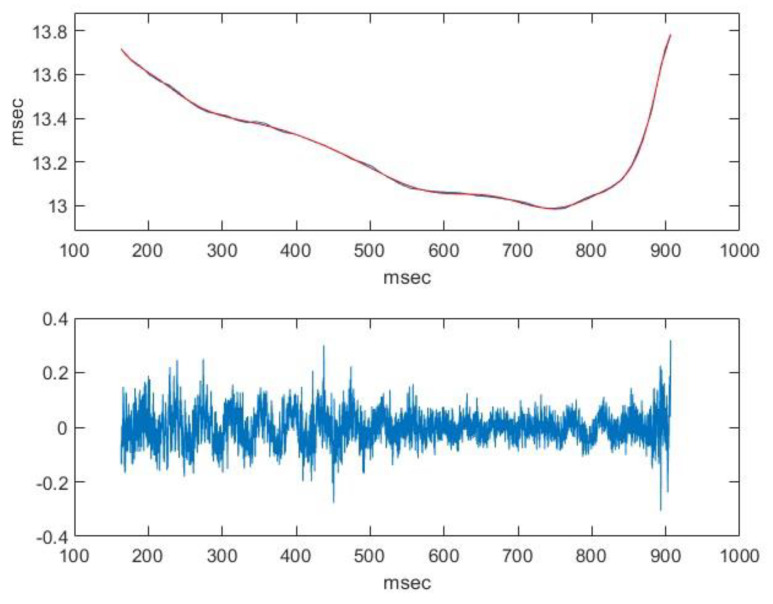
The function of the local period and the signal after the elimination of the quasi-periodic component. upper image: blue line—calculated values, red line—fitted polynomial.

**Figure 7 animals-14-00135-f007:**
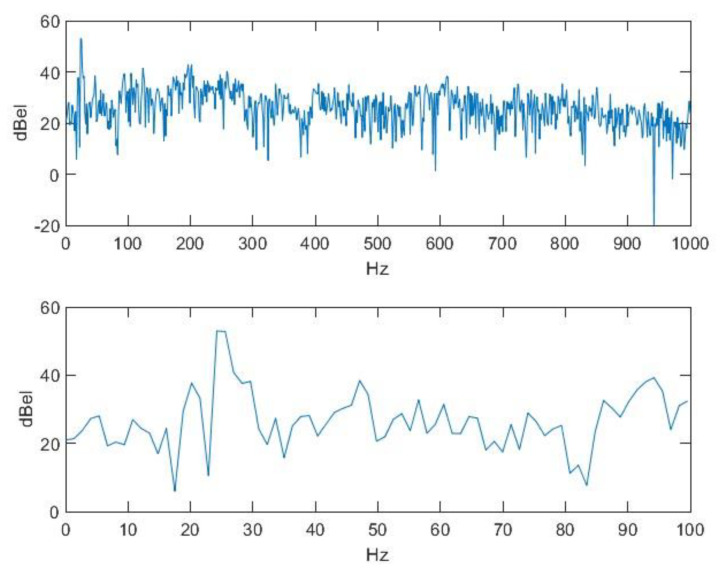
The energy spectrum of the difference signal and its magnified image.

**Figure 8 animals-14-00135-f008:**
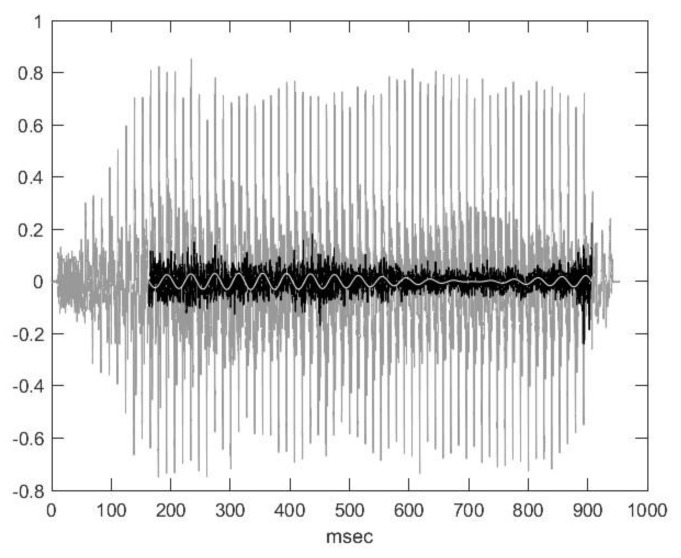
The three-time signals together. Original signal (grey) and the resolution of the signal in the lower half of Figure 6 is shown by a ‘low-pass (white) and high-pass (black) filter’ (the two ‘cutoff frequencies’ are identical).

**Figure 9 animals-14-00135-f009:**
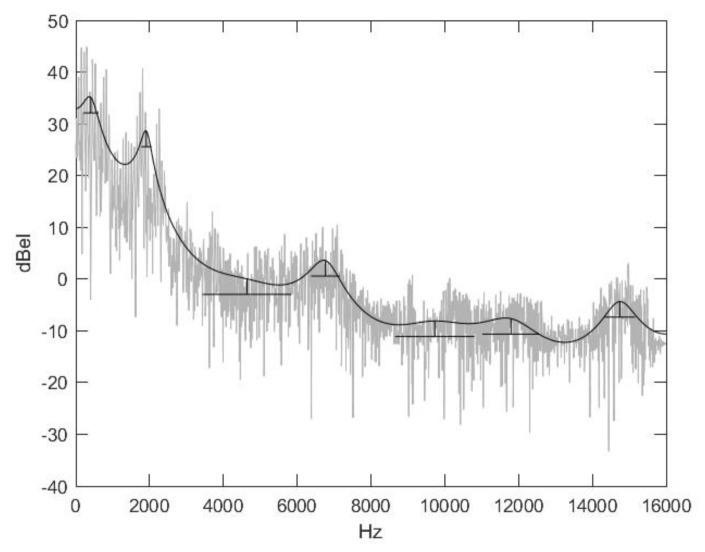
Formant structure of a section of the signal. Black line: formant structure; horizontal lines: formant’s width.

**Figure 10 animals-14-00135-f010:**
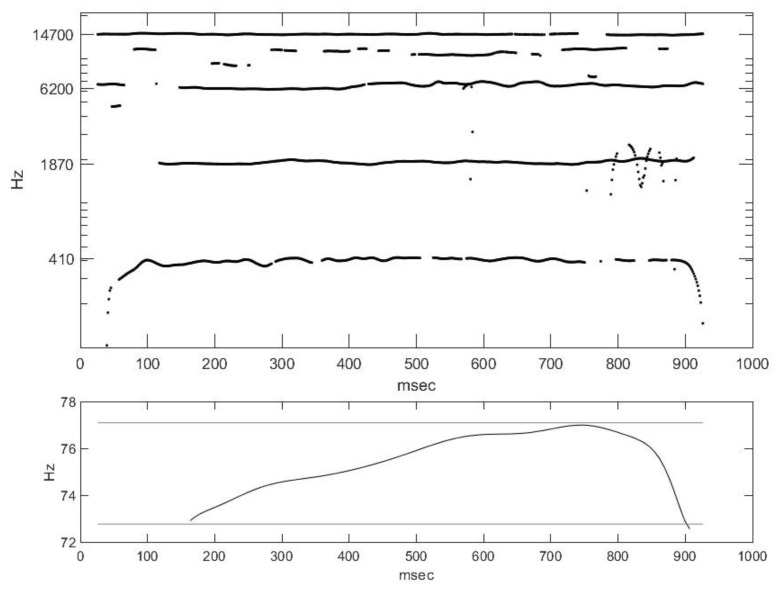
Temporal change in formant structures and the fundamental frequency.

**Table 1 animals-14-00135-t001:** Effect of maternal sound, day of test, and treatment sequence on heart rate, heart rate variables, and flight speed of 23 Hungarian grey cattle during restraint handling.

Trait	Maternal Sound		Day of Test		Sequence		*p*		
	No	Yes	1st	2nd	YN	NY	Sound	Round	Seq.
HR (1/min)	34.4	25.8	30.0	31.1	35.0	23.9	0.1544	0.7667	0.0852
HR max (1/min)	121.6	115.4	126.4	110.2	118.4	118.5	0.4824	0.1415	0.9833
SD1 (ms)	16.9	17.1	19.1	14.9	14.5	20.2	0.9131	0.2220	0.1057
SD2 (ms)	16.9	17.7	19.3	15.2	15.2	19.9	0.9484	0.2312	0.1647
SD2/SD1	0.97	1.07	1.03	1.02	1.04	1.01	0.0110	0.8051	0.5139
RMSSD (ms)	23.5	23.6	26.5	20.5	20.1	28.1	0.8955	0.2200	0.0979
pNN50 (%)	9.20	9.45	9.48	9.08	9.01	9.57	0.1997	0.1650	0.0396
VLF (%)	12.6	12.9	13.5	12.0	13.5	11.7	0.9993	0.5727	0.5058
HF (%)	18.9	19.3	17.4	21.0	19.0	19.2	0.9083	0.1820	0.9794
LF (%)	57.2	57.1	59.9	54.4	53.2	62.3	0.8428	0.1365	0.0176
LF/HF	3.27	2.98	3.54	2.74	2.77	3.59	0.7319	0.1856	0.1480
Flight speed (s/m)	0.92	1.08	1.07	0.94	1.07	0.90	0.0409	0.1557	0.0199

## Data Availability

The data presented in this study are available upon request from the corresponding author. The data are not publicly available due to the policy of the funder.

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
