# Peer review of "Calming Hungarian Grey Cattle in Headlocks Using Processed Nasal Vocalization of a Mother Cow"

_animals, 2023, doi:10.3390/ani14010135_

Round 1

Reviewer 1 Report

Comments and Suggestions for Authors

The Title, Simple Summary and Abstract all imply that cows in a head crate are calmer when played the sound of their own dam. In fact the sound played to all the animals tested was from a single cow, not a dam of any of these animals. In addition it was a sound created by removing the first and last periods of the sound, then repeating the mid-section indefinitely so that it would have sounded nothing like a call from a dam to its calf.

More than half the manuscript is taken up by analysis of the sound, but most of that is poorly described and does not appear relevant to the specific problem considered here, calming an animal while in a head crate. This sound analysis is ignored in the summary and abstract, possibly because the relevance is not clear. The sound analysis might belong in a separate paper for a different audience.

The significance of the results is marginal and depends on selectively choosing some test results as more important than others. It is not clear what "Seq" means because the sentence explaining it does not make sense. But this effect had the most significant results. It might be the difference between heard-the-sound vs did-not-hear-the-sound, but "seq" implies that the order makes a difference, and only half were in that order.

Overall, the results appear to have little value, and the sound itself appears to have been selected in an arbitrary manner, without any real basis, while the benefits, if any, of using the method are minimal.

Comments on the Quality of English Language

Some sentences are poorly constructed and the manuscript needs review by an English editor.

A particular problem is the use of the term "dam", which should be limited to describing the relationship of a cow to a specific calf. It has been used here to refer to any cow that happens to have a calf, even where the calf is not relevant to to this study. The animals studied here may have calved recently, but are just "cows" for the purpose of this report.

Author Response

Dear Reviewer,

Thank you very much for reviewing our manuscript. Please find below our detailed responses.

Sincerely;

Adam Lenner

Responses to reviewer 1

Thank you very much for reviewing our manuscript. Please find below our detailed responses.

The Title, Simple Summary and Abstract all imply that cows in a head crate are calmer when played the sound of their own dam. In fact, the sound played to all the animals tested was from a single cow, not a dam of any of these animals. In addition, it was a sound created by removing the first and last periods of the sound, then repeating the mid-section indefinitely so that it would have sounded nothing like a call from a dam to its calf.

Thank you for noticing the incorrect use of the word dam. We have changed that term to „mother cow”, also as to the request of the academic editor. As the sound was recorded in a raw form, it needed further processing. It was important to clean it from noises, select the active component and concentrate it to be more effective on the individuals. Therefore, it is not just removing the first and last part. It is a similar approach when we are preparing a plant extract. We could use the whole plant as well, but than we would need a much higher amount and we could not be sure about the concentration. We provided more explanation in the abstracts and into the introduction as well.

More than half the manuscript is taken up by analysis of the sound, but most of that is poorly described and does not appear relevant to the specific problem considered here, calming an animal while in a head crate. This sound analysis is ignored in the summary and abstract, possibly because the relevance is not clear. The sound analysis might belong in a separate paper for a different audience.

Thank you for your comment, we provided more information about sound analyses in abstracts and in introduction. The sound analyses section seems to be long because of the number of figures. Research was needed for the analysis of the acoustic structure of cattle sounds, so that the specific information they carry could be determined. This is why we analyzed in detail the given nasal sounds, and the information gathered can facilitate the effective use of sound technology in precision livestock farming in the future. Nowadays many research has multidisciplinary approach. That is why we see more publications with a number of authors, because none of them can be a master of every scientific field. It also has the consequence, that the publications in part not really understandable to a reader, but it is important to gain some knowledge about other research areas to be able to cooperate effectively.

The significance of the results is marginal and depends on selectively choosing some test results as more important than others. It is not clear what "Seq" means because the sentence explaining it does not make sense. But this effect had the most significant results. It might be the difference between heard-the-sound vs did-not-hear-the-sound, but "seq" implies that the order makes a difference, and only half were in that order.

Thank you to bringing our attention that we poorly described the sequence meaning, and data accidentally not provided in table 1, which has been inserted and the following explanation is given in lines 305-313:

As half of the mother cows was tested in the sequence of sound played and not played (YN) and the other group had the opposite order (NY), we had the chance to test that whether this sequence had a significant effect on any of the parameters. Quite interestingly pNN50 (P=0.0396), LF (P=0,0176) and flight speed (P=0.0199) showed a positive response. While pNN50 and flight speed shows that cows in YN sequence were calmer (suggesting a kind of more positive association with the head lock), LF ratio was higher in NY sequence cows (a possible signal of negative association to head lock and less effective sound treatment). These data controversial, needs further studies to understand, but shows that HRV indices are sensitive parameters.

Overall, the results appear to have little value, and the sound itself appears to have been selected in an arbitrary manner, without any real basis, while the benefits, if any, of using the method are minimal.

Cattle vocalization was analyzed from several approaches in the past, but the nasal sound type that we selected has not yet been analyzed with the methods that we applied. Other researchers analyzed oral vocalizations triggered by separation, estrus, or other kinds of stress (Kiley, 1972; Padilla de la Torre et al., 2015; Green et al., 2019). The sound/voice of a mother is relaxing to the offspring. Therefore, we hypothesized that the core part of such vocalization can have a calming effect in stress situations. Animal welfare is very important nowadays, and in order to keep animal production public acceptance we have to do every effort to show that we do everything for the wellbeing of animals.

Reviewer 2 Report

Comments and Suggestions for Authors

I love this paper!

You tested such a nice common-sense idea in normal cattle working conditions. This could be a simple and effective widely applicable calming method.

Here are some of my comments for you to take or leave. I am already very curious to see your responses.

Did the people driving the cattle and those putting on the heart rate monitors know which group the cow was assigned to or were they blind to the treatment? This is an important aspect to mention as the behaviour of the experimenters could have influenced the behaviour (both of people & cows).

The last individuals are expected to experience more stress. Did you look at an order effect within tests, from the first to the last individual? 

Give more information on the dam whose vocalisations were used: what was her rank position, years in the group, sociability, relatedness to other individuals, sex and age of calf she vocalised to, context in which she vocalised ("while licking it"): was it after a separation by the mother or by the calf, after birth... ?

Simple summary: specify research question.

Line 30: "This was proven with the heart rhythm (SD1/SD2) and the so-called “flight speed” tests": the word "proven" is too strong as a claim. Specify the exact results of HRV & speed tests.

Line 105 Material and methods: there were seven individuals that did not provide good data: describe briefly why; this is very useful information too.  

Line 175 Stats: describe variables and effects

Line 183. 3.1. Sound analysis: this is a lot of detailed information on one vocalisation sequence. If you provide the call (as a wav or mp3) & the detailed information as a supplementary material, the description of the call in the results section can be limited. I would prefer a brief section on a few of the call aspects with known functional relevance (eg low frequency etc...), with discussion focusing on the literature on the receiver or sender's associated emotions.  

Is anything known on the normal variation within and among individuals for this or similar call type in cattle or related species?  

Results after 288: discuss and explain the sequence effects. This is important information.  

Results of PNN50 (significant sequence effect) and SD2/SD1 (significant sound effect) differ. Why would this be? 

Line 293: check sentence. It is incomplete. 

Line 298: check double. 

Discuss the difference among the HRV-measures. Why would the data on SD2/SD1 as an index of sympathovagal balance be more responsive to the treatment? 

Author Response

Dear Reviewer,

Thank you very much for reviewing our manuscript, and especially your very positive opinion about it. Please find below our detailed responses.

Sincerely;

Adam Lenner

Responses to reviewer 2

Thank you very much for reviewing our manuscript, and especially your very positive opinion about it. Please find below our detailed responses.

Did the people driving the cattle and those putting on the heart rate monitors know which group the cow was assigned to or were they blind to the treatment? This is an important aspect to mention as the behaviour of the experimenters could have influenced the behaviour (both of people & cows).

The examined cows entered the corridor on their own; consequently, their order was random, the people driving the cattle did not influence the order. The supervisor of the experiment (AL) was putting on the heart rate monitor, and played the sound when it was required, therefore he was aware of the treatment. W completely restructured the materials and methods section is completely restructured, and in lines 142-145 we inserted details about that.

The last individuals are expected to experience more stress. Did you look at an order effect within tests, from the first to the last individual? 

We did not look at an order effect. As the herd size was more than two times the test group, even the last animal was not alone in the paddock. Therefore, we think that this effect was not important in our case.

Give more information on the dam whose vocalisations were used: what was her rank position, years in the group, sociability, relatedness to other individuals, sex and age of calf she vocalised to, context in which she vocalised ("while licking it"): was it after a separation by the mother or by the calf, after birth... ?

The cow making the call was 9 years old and was giving birth to its third calf at the time of the sound recording. The nasal sound was recorded on the first day after the calving, when the cow was smelling and licking her one-day old calf. The weaning of calves takes place when they are 8 months old, until then the cows are kept together with their calves. The cow was added to the group on 16 March 2015. During livestock management, she enters the corridor on her own, without being driven, at the 98/200 place on average (85;45;100;151;155;68;81). Among 189 individuals, the average distance between the horn tips is 90 cm, in the case of the individual n. ID3266718997 it is 107 cm. The presence of the horn and its size affect the position in the ranking (Woodbury, 1941), but we did not examine this specifically. There was no close degree of kinship on the maternal line between the animals involved in the research.

Simple summary: specify research question.

We completely rephrased the simple summary, and we defined our hypothesis like that: We hypothesized that by removing the noises, selecting the most representative part and creating a repeated sound it will have a calming effect on cattle restrained in head locks.

Line 30: "This was proven with the heart rhythm (SD1/SD2) and the so-called “flight speed” tests": the word "proven" is too strong as a claim. Specify the exact results of HRV & speed tests.

We rephrased the sentence like: SD2/SD1 ratio (0.97 vs. 1.07 for no sound and sound played animals, respectively, P=0.0110) and flight speed (0.92 vs. 1.08 s/m for no sound and sound played animals, respectively, P=0.0409) indicates that the sound treatment had a calming effect on the cow restrained. Kindly see in lines 43-45.

Line 105 Material and methods: there were seven individuals that did not provide good data: describe briefly why; this is very useful information too.  

Animals were not completely still in the head lock which may resulted in electrodes disposition, furthermore the thick hair may have blocked the signal detection even though ultrasound gel was used to improve connectivity, resulting in unanalyzable RR data. Kindly see in lines 176-179.

Line 175 Stats: describe variables and effects

Variables has been described in section 2.4 and 2.5 . The effect of sound hearing, test day and the sequence of change over design were tested. Kindly see in lines: 196-197.

Line 183. 3.1. Sound analysis: this is a lot of detailed information on one vocalisation sequence. If you provide the call (as a wav or mp3) & the detailed information as a supplementary material, the description of the call in the results section can be limited. I would prefer a brief section on a few of the call aspects with known functional relevance (eg low frequency etc...), with discussion focusing on the literature on the receiver or sender's associated emotions.  

The sound analyses section seems to be long because of the number of figures. Research was needed for the analysis of the acoustic structure of cattle sounds, so that the specific information they carry could be determined. This is why we analyzed in detail the given nasal sounds, and the information gathered can facilitate the effective use of sound technology in precision livestock farming in the future. Therefore, with your kind agreement we would keep this section in the main body of the manuscript. The sound file will be added as supplementary material.

Is anything known on the normal variation within and among individuals for this or similar call type in cattle or related species?  

We were not able to find information about individual variance. An early research (Kiley, 1972) suggested that in a herd of mixed breed beef and dairy cattle there were six distinct call types comprising different combinations of five syllables. This suggest that there is some individuality in vocalization, but it has to be in part common. However, early studies did not have the instruments to analyze sound as detailed as we could do it in our research. Our detailed presentation of sound analyses can contribute to the examination of individual variations.

Results after 288: discuss and explain the sequence effects. This is important information.  

We discussed and explained sequence results and effects in lines 326-334 as follows: As half of the mother cows was tested in the sequence of sound played and not played (YN) and the other group had the opposite order (NY), we had the chance to test that whether this sequence had a significant effect on any of the parameters. Quite interestingly pNN50 (P=0.0396), LF (P=0,0176) and flight speed (P=0.0199) showed a positive response. While pNN50 and flight speed shows that cows in YN sequence were calmer (suggesting a kind of more positive association with the head lock), LF ratio was higher in NY sequence cows (a possible signal of negative association to head lock and less effective sound treatment). These data controversial, needs further studies to understand, but shows that HRV indices are sensitive parameters.

Results of PNN50 (significant sequence effect) and SD2/SD1 (significant sound effect) differ. Why would this be? 

Although the HRV parameters are calculated from the same RR intervals, only a few of them correlates to an another one. No research reported correlation between pNN50 and SD2/SD1 ratio. It suggests that they behave differently. HRV analyses is much more sensitive than heart rate, and can pick up little differences in cardiovascular system.

Line 293: check sentence. It is incomplete. 

We corrected the sentence.

Line 298: check double. 

We have corrected the sentence.

Discuss the difference among the HRV-measures. Why would the data on SD2/SD1 as an index of sympathovagal balance be more responsive to the treatment?

We gave more background information in the introduction (lines 87-100). The discussion has been extended in lines 343-362.

Round 2

Reviewer 1 Report

Comments and Suggestions for Authors

The response of the authors to a previous review, and the changes made to the manuscript are satisfactory, although it is not clear whether a better (or worse) result would be obtained with totally different editing of the sound. However, the explanation give for the sound production is adequate.

I fully support the idea of a multidisciplinary approach, and am strongly in favour of methods to improve animal welfare, although it is not clear that the method proposed will actually be used to benefit other animals in the future. However, this study may lead to further work on the use of calming sounds during animal procedures.

Comments on the Quality of English Language

There has been some improvement in the quality of English, but most sections were completely rewritten, leaving many sentences poorly written in those rewritten sections. However, it was possible to understand the meaning, and the manuscript would be improved by a good editor.

Author Response

Dear  Reviewer,

Thank you very much for your precious time for assessing our manuscript second time.

Here are our answers;

Sincerely,

Adam Lenner

Reviewer 2 Report

Comments and Suggestions for Authors

I think the revision was nicely done. There are some minor corrections to be made. 

I still think the amount of detail on the acoustic analysis is a bit disproportionate for this paper, but the authors are quite attached to it for good reasons and who am I to make them move it to the supplementary materials. I guess the editor can take a look at it and see what he/she thinks.

I like the fact that the call will be provided as supplementary material.

3.2 line 315: "two test day" should be: two test days

line 319: "was" is missing

line 319: "distance" should be "distances"

line 336: "we had the chance to test 335 that whether this sequence had a significant effect on any of the parameters" should be: ""we had the chance to test 335 whether this sequence had a significant effect on any of the parameters" 

line 341: "These data controversial" should be "These data are controversial"

line 341: "needs further studies to understand" --> and further study is needed to understand... " Specify what exactly, the mechanisms? 

line 343: "Heart rate variability (HRV) parameters is very useful" should be "are very useful"

line 349: this sentence is incomplete & should be checked: " The researchers examined mainly the SDNN and RMSSD from the time domain, while LF, HF and LF/HF 350 ratio from the frequency domain (but various frequency ranges used)"

line 359: " There results indicate, that regardless of the stress source lower ratio can be expected." a lower ratio or lower ratios

line 365: "Currently, there is ongoing Analysis of the effects of cattle’s nasal and oral sounds is ongoing at the present time." --> ""Currently, analysis of the effects of cattle’s nasal and oral sounds is ongoing."

line 380: "We have been able to clearly demonstrate a cow’s nasal vocalization on cattle with both the SD2/SD1 parameters and flight speed measure." --> "We have been able to demonstrate an effect of hearing a cow’s nasal vocalization on the SD2/SD1 parameters and flight speed measures in cattle."

The sequence effect is also interesting. I like the addition of these data. 

I still remain a bit worried about the fact that the experimenters are not blind to the treatment. The differences in speed & HRV parameters are very small and may equally be caused by slightly biased differences in human treatment. It is however challenging to find a way to design a protocol that takes care of this. I guess future replication studies are needed to clarify this.

Comments on the Quality of English Language

There were some linguistic or editing errors in the revision. I think I found them all but a good second check would do no harm. 

Author Response

(The authors gave the same response as above.)
